# Molecular Signatures Integral to Natural Reprogramming in the Pigment Epithelium Cells after Retinal Detachment in *Pleurodeles waltl*

**DOI:** 10.3390/ijms242316940

**Published:** 2023-11-29

**Authors:** Yuliya Markitantova, Alexander Fokin, Dmitry Boguslavsky, Vladimir Simirskii, Aleksey Kulikov

**Affiliations:** Koltsov Institute of Developmental Biology, Russian Academy of Sciences, 119334 Moscow, Russia; yuliya.mark@gmail.com (Y.M.); a.m.kulikov@idbras.ru (A.K.)

**Keywords:** adult newt, retinal detachment, regeneration, retinal pigment epithelium (RPE), natural RPE reprogramming, dedifferentiation, transcriptome, endogenous defense system, EMT, DEGs, biological processes, molecular pathways

## Abstract

The reprogramming of retinal pigment epithelium (RPE) cells into retinal cells (transdifferentiation) lies in the bases of retinal regeneration in several Urodela. The identification of the key genes involved in this process helps with looking for approaches to the prevention and treatment of RPE-related degenerative diseases of the human retina. The purpose of our study was to examine the transcriptome changes at initial stages of RPE cell reprogramming in adult newt *Pleurodeles waltl*. RPE was isolated from the eye samples of day 0, 4, and 7 after experimental surgical detachment of the neural retina and was used for a de novo transcriptome assembly through the RNA-Seq method. A total of 1019 transcripts corresponding to the differently expressed genes have been revealed in silico: the 83 increased the expression at an early stage, and 168 increased the expression at a late stage of RPE reprogramming. We have identified up-regulation of classical early response genes, chaperones and co-chaperones, genes involved in the regulation of protein biosynthesis, suppressors of oncogenes, and EMT-related genes. We revealed the growth in the proportion of down-regulated ribosomal and translation-associated genes. Our findings contribute to revealing the molecular mechanism of RPE reprogramming in Urodela.

## 1. Introduction

The regeneration of neural tissues has for a long time remained a priority area in biomedical research [1,2,3,4,5,6,7]. Salamanders (Urodela) are the most regeneration-competent animals. Certain salamanders are able to complete structural and functional recovery of the eye retina after various types of injury after disconnections of the retinal pigment epithelium (RPE) and neural retina (NR). The differentiated RPE cells, and also the cells of the growth ciliary marginal zone on the periphery of the retina, are the source of a new fully regenerated retina in newts [8,9,10,11,12,13]. Retinal regeneration is a multistage process. Early molecular genetic- and cellular processes are primarily aimed at protecting cells from cellular stress, inflammation, and degenerative changes and are regulated at the local and organismal level [2,11,14]. The transdifferentiation of RPE cells (in vivo natural reprogramming) into retinal neurons and glia is the key process of retinal regeneration in adult Urodela as a response to RPE detachment from the photoreceptor cell layer. The main cellular events during RPE cell reprogramming are as follows: the release of RPE cells from the layer, loss of the original phenotypic features (dedifferentiation), and formation of the unique transient population of multipotent neuroblasts. The time needed for RPE cells to acquire a new differentiation was estimated by morphological criteria as the time of cell exit from the reproduction cycle. Proliferating neuroblasts form the NR primordium: after 6–7 cycles of divisions, they exit the reproduction cycle and then acquire the phenotypes of retinal neurons and glial cells. A population of RPE cells is preserved in the layer, and these cells proliferate and restore the RPE layer and its functional connections with a retina that is regenerated de novo [9,15].

Unlike mammals, Urodela regenerate their retina without epithelial mesenchymal transition (EMT) and scar formation. Cellular reactions to retina damage as a result of impaired intercellular connections between the RPE and the NR are common in vertebrates. But despite the similarity, the cellular reactions to injury and stress lead to opposite final results in mammals [16]. Disorders in communication between the RPE and retinal photoreceptors activate the proliferation and conversion of RPE cells into the path of EMT in humans [2,17]. RPE cell conversion along the mesenchymal pathway (EMT) is considered to be among the major causes of blindness in humans, including proliferative vitreoretinopathy (PVR) and age-related macular degeneration [16,18,19]. For example, as a complication of rhegmatogenous retinal detachment, RPE cells die, and some cells, with the acquisition of myofibroblast properties, together with glial cells, form fibrotic epiretinal membrane, which is the main cause of PVR and other eye pathologies associated with fibrosis triggering the neurodegenerative processes [2,20].

Studies on RPE cell reprogramming in the newt are stimulated by the assumption that understanding the reprogramming mechanisms will pave the way for the creation of new regenerative technologies for the treatment of RPE-related retinal disorders and for restoring the retinal functions that are lost as a result of disease or injury in mammals [21,22]. The ability of RPE cells to change their specialization and become retinal neurons and glial cells in Urodela (thus realizing their retinogenic potential) makes it possible to search for the key molecular factors of initial competence, allowing us to overcome EMT and cellular stress and to achieve a successful implementation of the RPE reprogramming process into neurons in vivo. Studies on newt tissue regeneration are making progress thanks to incorporating highly efficient technologies and allowing us to use microquantities of tissue for the analysis of transcriptomes, genomes, and proteomes [23,24], as well as for the analysis of gene functions [25,26,27]. The application of these approaches allow us to unravel the cellular and molecular mechanisms of aging and its relations with regeneration [28,29].

The construction of in silico transcriptomes via RNAseq/de novo assembly from different stages of regeneration is an effective approach, as the transcriptomic data sets are highly informative for bench-screening the candidate genes, their targets, and functional studies. To our knowledge, there is only one study examining the transcriptomes of the early processes of retinal regeneration in the adult newt *Cynops pyrrhogaster* [22]. In that study, models of total surgical removal of the NR were used to induce retinal regeneration, and the transcriptome data were obtained from the total RNA from different tissues of the eyeball samples taken on day 0–14 (0, 10, 14) [22].

In the present study, we have focused on transcriptome analysis of the initial stages of RPE cell reprogramming, using a model of experimental mechanical retinal detachment in *Pleurodeles waltl* (Figure 1). The morphological stages of RPE reprogramming have been described previously [30]. Special attention was paid to the period of RPE cell reprogramming preceding the emergence of the first progenitor cells (neuroblasts) in order to elucidate the mechanisms that contribute to this unique phenomenon. The transcriptome analysis was based on a comparison with annotated databases. This approach allowed us to highlight the potential key genes and their GO-cellular and metabolic pathways, which are involved in the specific gene programs and biological processes that drive and support RPE reprogramming in tailed amphibians.

## 2. Results

### 2.1. Similarities and Differences in Gene Expression in Normal RPE Cells and at the Early and Late Stages of RPE Regeneration

Gene expression profiling reveals the regeneration-specific molecular activities of RPE cells during reprogramming after the retinal detachment (4 days and 7 days po). A total of 1003 genes exhibited significant differential expression at the studied time points during regeneration. Of these genes, 228 changed their expression by four times or more, either up or down. A heat map, comparing the gene expression patterns in RPE cells from the normal retina and RPE at different stages of retinal regeneration, was built using this sample of genes (Figure 2).

Both genes and experimental samples were hierarchically grouped into functional clusters based on the similarity of their expression patterns. The gene expression pattern in RPE cells at the early stage of regeneration was closer to normal tissue than to RPE at the later stage of reprogramming. The differences between the “early”, “late”, and “norm” classes are characterized by an increased expression of genes in the upper and lower thirds of the cluster in the “early” class and a decreased expression in these parts of the cluster in the “late” class. The opposite picture is observed in the “norm” group, which is characterized by an increased expression of genes in the central part of the cluster.

### 2.2. Changes in Gene Expression Activity in Retinal Cells at Different Stages of RPE Cell Reprogramming

Significant changes in gene expression during RPE reprogramming are characteristic of both the early and late stages (Figure 3a,b).

It should be noted that we discuss the differential gene activity when we use the term RNA quantity (transcripts), and we use the term proteins when we discuss the functional activity. Both of the experimental versions demonstrate a similar pattern in expression activity changes in regenerating tissues compared with normal RPE tissue. A third of these genes decrease their expression, and a third increase their expression. The majority of differentially expressed genes (DEGs) have a low level of expression in normal RPE and transdifferentiating RPE cells and a significant, but relatively small, change in expression (2–3-fold). The similar low changes in expression are also characteristic of genes with high levels of average expression activity (right side of the distribution are shown in the diagrams (Figure 3a,b)). In both versions of the experiment, a group of genes were identified that have expression rates close to zero in normal RPE and increase their expression many times in the transdifferentiating RPE cells (line is represented by outliers in the charts in the upper left corner).

Despite the apparent similarity of Ma plots, a detailed pattern of gene expression in samples of different stages of RPE cell transdifferentiation revealed significant differences, in accordance with the heat map data (Figure 2). A Venn diagram evaluation demonstrates the distribution of unique and common genes with up- and down-regulated expression in RPE cells at the early and late stages of reprogramming (Figure 4). For diagram construction, the 1003 genes whose differential expression was confirmed by FDR indicators at the *p* < 0.05 level were selected (Appendix A). The most noticeable is the one-and-a-half to two-fold predominance of genes with differential expression at a late stage of RPE reprogramming: the 441 genes versus 250 for up-regulated genes and 346 genes versus 264 for down-regulated genes. Between 50% and 60% of genes that are differentially expressed at the early stage of RPE reprogramming retain in an altered expression state at a late stage.

The sets of DEGs for the early and late stages of regeneration show an intersection of scores of sets that are common to both stages. They make up from half to two-thirds of the genes that change their expression at the early stage of regeneration. Approximately half and one-third of up- and down-regulated genes, respectively, restore the normal expression at a later stage of RPE reprogramming. DEGs that are common to both stages generally maintain a constant up- or down-regulated level.

However, the nine up-regulated genes increase their expression two-fold or more upon transition to a later stage of regeneration. For down-regulated genes, the 10 genes whose expression continued to decrease by two or more times at a later stage were noted, as were the 12 genes that increased their expression to the same extent. The total number of unique genes in both up- and down-regulated sets is 1019 (and not 1003). This is due to the small number of genes that changed their pattern of differential expression from up-regulated to down-regulated, and vice versa, which were simultaneously included in both sets. The 13 up-regulated genes at the early stage of regeneration, upon transition to a later stage, exhibited a down-regulated expression compared with the norm, and 3 genes changed their expression profile from down-regulated at the early stage of regeneration to up-regulated (Appendix A). We have annotated the 10 genes from this set (Appendix A), including the FRIHB gene, which continued to decrease their transcription at the late stage and were involved in the transport of iron ions. This group also included the seven genes that exhibited increased expression at the early stages of retina regeneration. Their expression continued to increase significantly at the late stage of RPE reprogramming during retinal regeneration. Five of them are known to be important regulators of proliferation and apoptosis. 

### 2.3. Up-Regulated Genes at the Studied Stages of the RPE Reprogramming 

Among the annotated transcripts, 83 significantly increased their expression at an early stage of RPE reprogramming, and 174 at a late stage (Appendix A). Further analysis of the protein classes and the ontological processes and functions associated with them was carried out using filtered data. The transcripts that have high homology with two or more structurally and functionally different proteins were excluded from the analysis. The exceptions were the transcripts that had homology in the set of different protein variants obtained for the genomes of amphibians or fish. In these cases, the only “correct” homology was taken to be a protein/gene from the corresponding amphibian or fish species. The final set of up-regulated analyzed genes included the 83 genes from the samples of the early stage of regeneration and the 168 genes from the samples of the late stage. A larger list of DEGs from RPE cells yielded a range of biological process annotations: these DEGs were statistically associated with the main GO terms. 

We used the bioinformatics tool DAVID [31] for the conversion ID of evolutionary orthologs into human gene ID. The analysis was carried out in the Panther Classification System, using the most representative human GO libraries. The results of the functional activity analysis of up-regulated proteins are presented in absolute values of their number in the classes (Figure 5). The size of most groups is noticeably higher at a later stage of RPE reprogramming. This is essential to determine the significance of the observed process enrichments. The processes associated with catalytic activity and binding activity are noticeably enriched at both stages of regeneration. However, the significance of the enrichment of catalytic activity processes with overexpressed genes is not confirmed in both cases. This is due to the large number of genes from this category (5640) and also to the closeness of the expected and obtained numbers of up-regulated genes in the samples. The binding activity is associated with mRNA binding processes that determine the specificity of the processing, splicing, and translation of mRNA (“early” sample: *p*-value 1.91 × 10^−^^12^, FDR 9.66 × 10^−^^9^ and “late” sample: *p*-value 9.45 × 10^−^^29^, FDR 4.78 × 10^−^^25^). The significance of the enrichment of up-regulated genes/proteins of all other classes at the early stage of RPE reprogramming has not been confirmed while taking into account the FDR values.

For the late stage of RPE reprogramming, significant enrichment was also shown for the structural molecule activity (*p* 6.73 × 10^−^^6^, FDR 0.003), including actin filament binding (*p* 5.59 × 10^−^^5^, FDR 0.019), cell adhesion molecule binding (*p* 1.46 × 10^−^^9^, FDR 1.23 × 10^−^^6^), unfolded protein binding (*p* 1.27 × 10^−^^4^, FDR 0.036), translation regulator activity (*p* 7.55 × 10^−^^5^, FDR 0.024), and primary active transmembrane transporter activity (*p* 8.13 × 10^−^^5^, FDR 0.024)

The analysis of the protein classes’ enrichment demonstrates an increase in processes that are associated with the general expression activity of genes, primarily determining the structure of the cell cytoskeleton, the cell’s response to stress, and intercellular contacts at a late stage of RPE reprogramming. The transmembrane transporter activity is associated with ATP synthesis and increased respiratory activity of cells, which explains the increase in metabolic activity. 

The use of the protein classification “PANTHER GO-Slim Molecular Function” to analyze the functional activity of proteins has allowed us to obtain a more detailed picture (Appendix A), which is not fundamentally different from that discussed above. 

The data demonstrate the influence of the Hsp90 family chaperone increase on the cellular stress response; the increased amino-acid kinase and phosphatase inhibitor classes influence the signaling and metabolic cascades activity; the increased P53-like transcription factor and HMG box transcription factor classes influence the transcriptional activity changes in the RPE genome during cell reprogramming.

The enrichment of biological process analysis for the up-regulated genes/proteins demonstrates the activation of processes of apoptosis and genomic instability, cell differentiation, and migration. A significant increase in up-regulated genes at an early stage of RPE reprogramming (Figure 6a) was shown for proteins of the actin-myosin complex, including cytoskeletal proteins, translation processes, post-translational modification, secretion of proteins, and proteins of the stress response to DNA damage stimulus. These processes are subject to significant development and enrichment with overexpressed proteins, including the regulators assembling actomyosin complexes, triggers and participants of the apoptotic signaling pathways, and translation activators, at the late stage of RPE reprogramming (Figure 6b). The increase in the cytoplasmic response to cellular stress is associated with an increase in up-regulated proteins, which are involved in post-translational modification and vesicular trafficking. The processes associated with the viral genome can indicate genomic instability in RPE cells and, as a result, an increase in the frequency of transpositions of mobile elements.

The analysis of the cellular localization of up-regulated proteins summarizes the patterns described above. At the early stage of RPE cell reprogramming, significant enrichments were shown for the proteins of the mitochondrial respiratory chain complex (*p* 4.66 × 10^−^^4^, FDR 0.035), contractile fiber (*p* 9.17 × 10^−^^5^, FDR 0.010), intracellular organelle lumen (*p* 2.80 × 10^−^^5^, FDR 0.004), vesicle (*p* 1.69 × 10^−^^6^, FDR 3.04 × 10^−^^4^), and extracellular exosome (*p* 4.82 × 10^−^^8^, FDR 4.77 × 10^−^^5^). This picture indicates an increased cellular respiration, changes in the cell cytoskeleton, activation of vesicular traffic, and protein secretion. At a later stage of RPE reprogramming, the up-regulated proteins show significant enrichment for all localization characteristics, which are described for the early stage. In addition, the late stage was enriched by the cytoplasmic stress granules (*p* 1.52 × 10^−^^7^, FDR 1.88 × 10^−^^5^), focal adhesion complexes (*p* 1.31 × 10^−^^5^, FDR 0.001), smooth endoplasmic reticulum (ER) (*p* 7.15 × 10^−^^5^, FDR 0.005), polysomes (*p* 9.97 × 10^−^^4^, FDR 0.038), nucleus (*p* 4.99 × 10^−^^5^, FDR 0.003), perinuclear region of cytoplasm (*p* 6.89 × 10^−^^4^, FDR 0.028), and cell junction complexes (*p* 1.54 × 10^−^^6^, FDR 1.33 × 10^−^^4^). For a more detailed picture of the regeneration processes related to the increased transcriptional and translational activity of genes, the remodeling of the cytoskeleton involving actomyosin complexes, changes in intercellular contacts, further strengthening of vesicular traffic, and protein secretion have been demonstrated. It should be noted that up-regulated genes are associated with translational activity (Appendix A) and represented mainly by Poly(U)-binding-splicing factors, elongation factors, pre-mRNA-splicing factors, nuclear cap-binding proteins, splicing factors, and other proteins that determine the maturation and transport of RNA to the ribosome.

### 2.4. Down-Regulated Genes at the Studied Stages of the RPE Reprogramming Process

In the set of annotated transcripts, 46 had a significantly reduced expression level at the early stage of RPE reprogramming, and 61 transcripts had a significantly reduced expression at the late stage (Appendix A). The analysis of biological processes, marked by these transcripts, was carried out in the same way as for the up-regulated transcripts. As a result, we have obtained a set of annotated transcripts, including genes/proteins, that represented the human evolutionary orthologues. Forty-four transcipts were down-regulated at the early stage of regeneration, and fifty-two were down-regulated at the late stage of RPE reprogramming.

The analysis of the functional activity of down-regulated proteins demonstrates a decrease in their number in the transporter activity and catalytic activity classes at a late stage of RPE reprogramming (Figure 7).

The relatively high numbers of down-regulated proteins in the classes of the structural molecule activity and binding, similar to the increase in the number of such proteins at a later stage of RPE reprogramming, are associated with the same group of ribosomal structural proteins that bind to ribosomal RNA (Appendix A).

The significance of enrichments that are common to both stages of RPE reprogramming was confirmed for the classes of structural constituents of ribosome (earlier stage: *p* 1.70 × 10^−^^19^, FDR 8.62 × 10^−^^16^; late stage: *p* 5.07 × 10^−^^28^, FDR 9.66 × 10^−^^9^), RNA binding (earlier stage: *p* 6.02 × 10^−^^15^, FDR 1.52 × 10^−^^11^; late stage: *p* 1.97 × 10^−^^10^, FDR 1.99 × 10^−^^7^), and cell adhesion molecule binding (earlier stage: *p* 1.51 × 10^−^^5^, FDR 0.009; late stage: *p* 1.85 × 10^−^^9^, FDR 9.35 × 10^−^^6^). For the late stage, the significance of the enrichment of down-regulated ribosomal proteins from the group exhibiting ubiquitin ligase inhibitor activity (*p* 1.27 × 10^−^^8^, FDR 9.16 × 10^−^^6^) within the classes’ molecular function regulator activity and catalytic activity have also been shown.

The picture we have obtained using the “PANTHER GO-Slim Molecular Function” classification showed the placement of all down-regulated ribosomal proteins in the translational protein class and the weak insignificant enrichment of all other functional protein classes (Appendix A).

The analysis of biological process enrichment associated with down-regulated genes/proteins demonstrates a weak influence of genes with reduced expression activity on translation processes at both stages of RPE reprogramming and with cell adhesion at a late stage (Figure 8). It can be assumed that there is a decrease in the intensity of cell contacts and an ambiguous effect on the activity of protein translation. In this case, the translation efficiency depends on the decreasing rate of the assembled ribosomal complexes that are associated with a decrease in the expression activity of genes that encode ribosomal proteins. The translation efficiency depends simultaneously on the increase in the selective ubiquitination and Neddylation of proteins under the influence of the ubiquitin ligase inhibitor activity of the same ribosomal proteins.

The enrichment of the signaling cascades of up- and down-regulated proteins was also assessed. Up-regulated genes/proteins that significantly enriched the signaling cascades are associated with the formation of conditions for protein processing in ER, folding and muscle contraction, regulation of spliceosome and ribosome assembly, initiation of translation and apoptosis, regulation of transposon activity, cellular respiration and response to stress, and immune response (Appendix A).

Among the down-regulated genes/proteins, there is a significant number of ribosomal proteins and proteins that regulate ribosome assembly, which in the Reactom Data Base are expressed as many signaling cascades that include ribosome assembly as an integral part of a broader process. In all cases where these processes showed enrichment predominantly or exclusively at the expense of ribosomal proteins, we assessed them as signaling pathways of rRNA processing and translation. In other cases, a significant enrichment by down-regulated genes can be noted, i.e., decreased activity of termination serine/threonine (O)-linked glycans biosynthesis pathways (involving the GalNAc-peptide linkage), TNF signaling, C-type Lectin receptor signaling, and VEGFA VEGFR2 signaling (Appendix A).

As expected, cellular localization is significantly enriched for ribosomes (earlier stage: *p* 2.40 × 10^−^^19^, FDR 1.58 × 10^−^^16^; late stage: *p* 5.07 × 10^−^^28^, FDR 2.57 × 10^−^^24^), including the large subunit (earlier stage: *p* 4.35 × 10^−^^18^, FDR 2.15 × 10^−^^15^; late stage: *p* 7.40 × 10^−^^21^, FDR 2.44 × 10^−^^18^) and small subunit (earlier stage: *p* 7.18 × 10^−^^9^, FDR 2.03 × 10^−^^6^; late stage: *p* 6.63 × 10^−^^10^, FDR 1.09 × 10^−^^7^), small subunit processome (earlier stage: *p* 2.01 × 10^−^^4^, FDR 0.012; late stage: *p* 6.56 × 10^−^^7^, FDR 4.64 × 10^−^^5^), nucleus (earlier stage: *p* 6.75 × 10^−^^4^, FDR 0.038; late stage: *p* 1.64 × 10^−^^9^, FDR 2.51 × 10^−^^7^), extracellular layer components (late stage: *p* 4.43 × 10^−^^5^, FDR 0.002), focal adhesion (earlier stage: *p* 2.78 × 10^−^^6^, FDR 3.45 × 10^−^^4^; late stage: *p* 1.96 × 10^−^^8^, FDR 1.85 × 10^−^^6^), and extracellular exosome (earlier stage: *p* 6.64 × 10^−^^7^, FDR 1.32 × 10^−^^4^; late stage: *p* 2.42 × 10^−^^9^, FDR 3.20 × 10^−^^7^). In the latter case, the described cellular localization is predominantly also associated with ribosomal proteins.

## 3. Discussion

### 3.1. Molecular Characteristics of the Early Stages of RPE Reprogramming

We have performed transcriptome analysis of DEGs and signaling pathways that are specific to RPE cell reprogramming in the *P. waltl*. Selected annotated genes from the DEGs category in the early and later stages of RPE reprogramming were grouped according to the primary in vivo function determined for their homologs (Appendix A). To interpret the biological significance of the DEGs changes, we used an integrated bioinformatics analysis that expanded on traditional analysis methods, such as GO and pathway analysis. The results of the present study show that early transcriptional changes during RPE cell reprogramming, which is compared with the norm, includes genes whose functions are associated with a number of key biological processes, including cellular metabolism, transcription, extracellular matrix (ECM), cytoskeleton, cell–cell contact remodeling, cell cycle, apoptosis, chromatin structure and remodeling, RNA translation and protein processing, “protein localization to ER, “translational initiation”, “cytoplasmic translation”, post-translational modification, etc. The GO analysis confirms the data and statistically highlights several genes that contribute to cell reorganization. We have discovered a significant enrichment of terms that are related to cytoskeletal reorganization and up-regulation of the genes encoding the proteins that are involved in endogenic defense systems in response to oxidative stress (OS). The analysis also demonstrates that only a small number of terms that are related to developmental processes are enriched for those genes that exhibit two-fold up-regulation (Appendix A).

In this section, we highlight several genes from different functional clusters in the context of closely interconnected molecular and cellular processes of RPE conversion, taking into account their multifunctionality. It is important to note that despite the evolutionary conservatism of functions for homologous genes, taxon-specific features are possible.

### 3.2. Changes in the Biosynthetic Activity

We have obtained data indicating that the process of RPE reprogramming in newts *P. waltl* is accompanied by changes in the transcriptome pattern that is related to regulating the activity of DNA, RNA, and general protein synthesis, maintenance of genomic integrity, remodeling of specific syntheses, and an appearance of specific proteins. It should be emphasized that several genes among DEGs encode the regulatory proteins. We have found a significant increase in the transcription of the genes *FUS*, splicing factors *SF3A3*, *SFPQ*, *CTR9 Paf1/RNA Polymerase II Complex Component*, *PHF12*, *PUM*, *PABP1*, *THRAP3*, *ATP6*, *RADI*, *RACK1* et al. at 4 days of RPE reprogramming. The *FUS* gene encodes nucleoprotein FUS, which belongs to the FET family of RNA-binding proteins. FUS functions include the regulation of the DNA and RNA metabolism, including DNA repair, transcription, RNA splicing, maintenance of genomic integrity, and mRNA/microRNA processing [32]. There is also evidence that FUS is responsible for some of the pathology seen in neurodegenerative diseases [33]. *PHF12* (*PF1*) encodes PHD Finger Protein 12, which takes part in maintaining the equilibrium of the rRNA processing pathway and ribosome biogenesis. PHF12 is involved in the negative regulation of transcription EMT and maintenance of stem cells [34]. PUM1 (RNA-binding protein PUMILIO1) protein is known to regulate the transcription of proteins that take part in cell adhesion, cell cycle regulation, maintenance of stem cells, regulation of the genes involved in innate immunity, and genome stability. [35,36]. PUM1 directly regulates the neurodegenerative-related protein ATAXIN1 and negatively regulates gene expression by repressing the translation of mRNAs to which they bind [37]. *PABP1* encodes the Polyadenylate-binding protein 1 (PABPN1) located in the nucleus, where it binds to the poly(A) tails of pre-mRNAs and facilitates their nuclear-cytoplasmic transport, translation, and stability. PABP1 binding is required for its nuclear-mediated degradation and function regulation [38]. PABPC1 binds to the poly(A) tail and interacts with eIF4G, forming the structure that stabilizes the circularization of mRNAs. This structure is required for the prevention of mRNA degradation via nonsense-mediated mRNA decay (NMD) [39,40]. *THRAP3* encodes TR150 protein that is involved in the response to DNA damage. TR150 regulates the nuclear-transcribed mRNA catabolic process, RNA metabolism, and DNA repair and stability. TR150 enables phosphoprotein binding activity, such as transcription coactivator activity, by positively regulating the transcription by RNA polymerase. TR150 takes part in the regulation of cyclin-D1/CCND1 mRNA stability, probably by acting as a component of the SNARP complex, which associates with both the 3′end of the *CCND1* gene and its mRNA (PubMed: 24043798) (TR150_HUMAN, Q9Y2W1). *RACK1* encodes the protein coding receptor for activated C kinase 1 and is a key mediator of numerous pathways. RACK1 interacts with the ribosomal machinery and plays a significant role in shuttling proteins around the cell, anchoring proteins at particular locations, and in stabilizing protein activity. RACK1 recruits ribosomes, and PKC and scaffolds are central components of the MAPK and the PI3K pathways [41]. RACK1A functions in innate immunity by interacting with the GTP form of Rac1 and plays a central role in the production of reactive oxygen species (ROS) in response to damage [42]. RACK1 regulates cell proliferation, adhesion, and migration [43], remodeling the components of the cell cytoskeleton, which are all essential components of cell transformation [44]. RACK1 may, in addition, protect cells from apoptosis in cancer [45]. The data we have obtained from the analysis of RPE transcriptome are in good agreement with previously obtained results at the level of autoradiography and immunochemistry [8,46,47,48]. It was demonstrated on axolotl limb regeneration that rapid activation of protein synthesis that is sensitive to amino acid transport is a unique feature of the early regenerative response in Urodela [49]. The biosynthetic activity of the RPE cells ensures their preparation and implementation for the neuroblast formation and proliferation.

### 3.3. Remodeling Proteins of the ECM and the Surface of RPE Cells

The retinal regeneration is accompanied by changes in ECM and cell contacts early in the process of their reprogramming [50]. Transcriptome analysis revealed the transcriptional activation of the genes *ADT2*, *PLEC*, *CAPG*, *CDC42*, *LGMN*, *Septin-8*, *NOL8* et al., at the early stage of RPE reprogramming, the functions of which may be associated with cytoskeletal restructuring and ECM remodeling. *ADT2* encodes a member of the ADAMTS family of secreted proteinases that is related to metalloproteinases. ADAMS and ADAMTs are involved in cytokine processing and growth factor receptor shedding [51], modulate the extracellular signals, and remodel the ECM [52]. *PLEC* encodes a high-molecular-weight cytoskeletal linker protein plectin, which is a highly conserved and ubiquitously expressed intermediate filament-linking protein [53]. The *CAPG* gene (capping actin protein, gelsolin-like gene) encodes an actin regulatory protein, a member of the gelsolin/villin family of actin-regulatory proteins [54]. CAPG plays an important role both in the immune response and tumor progression [55]. *SEPT8* encodes protein Septin-8 (denoted SEPT), a member of the septin family of GTP-binding proteins (GTPases) that assemble as filamentous scaffolds [56]. They regulate several cellular processes (cytokinesis, membrane remodeling and compartmentalization, cytoskeleton rearrangement, vesicle trafficking, apoptosis, etc.) [57] and the release of neurotransmitters [58,59]. The balance of the SEPT8 transcript variant expression and their changes have been suggested to modulate the β-amyloidogenic processing through a mechanism affecting the intracellular sorting and accumulation of BACE1 [60]. Cdc42 regulates the organization of the actin and septin cytoskeletons and polarized secretion, directed by intracellular or extracellular spatial cues, such as cell–cell contacts or chemo attractants, via downstream PAK effectors [61]. *LGMN* encodes the asparaginyl-specific cysteine protease, a glycoprotein that is a member of the peptidase family C13 (Legumain). Its activity is a part of a specific intracellular stress response and could lead to catabolic metabolism of the main components of ECM, such as fibrin and collagen I, by activating the MMP-2/transforming growth factor-β1 signal [62]. LGMN function is also critical in the progress of tumors by regulating the cell microenvironment such as fibronectin [63]. The increase in *NOL8* transcription, which encodes Ras-related GTP-binding protein (MIM 608267) and take part in signaling, promotes cell proliferation, and growth was detected [64]. It was shown at the level of immunochemistry that RPE reprogramming is accompanied by changes in the patterns of the proteins, fibronectin, laminin, heparan sulfate proteoglycan, N-CAM, integrin beta1 et al., which ensure the stability of cell differentiation in the normal sample. The remodeling of cell–cell contacts, ECM components, and cytoskeleton allows RPE cells to move out of the layer and then to dedifferentiate and change their initial morphology. These changes occur under the influence of “permissive” signals of Fgf, Wnt, IGF1, mTOR, etc., from the cellular microenvironment and are under epigenetic regulation at the early stages of RPE reprogramming and lead to a change in the RPE cell phenotype into the retinal cell types [65,66,67,68,69,70].

### 3.4. Immune Response Genes, Protoncogenes, and Tumor Suppressors

The list of genes from the group of the early immune response, protooncogenes and tumor suppressors, includes the following: *B2MG*, *ZN462*, *PAWR*, *PCYT1A*, *DEK*, *MAL*, *RWDD1*, *PTEN*, *COX1*, *RAB1A*, *ILF2*, and *p53*, which have been observed at 7 days of RPE reprogramming. *B2MG* encodes Beta-2-microglobulin, which is a serum protein found in association with the major histocompatibility complex (MHC) class I heavy chain tumor marker [71]. COX1 (Mitochondrial cytochrome c oxidase I) makes an important contribution to inflammatory responses and in cytochrome-c oxidase activity. COX1 is predicted to be involved in electron transport, coupled proton transport, and mitochondrial electron transport. *ZN462* encodes protein, which takes part in the regulation of pluripotency and the differentiation of the embryonic stem cells by targeting SOX2, POU5F1/OCT4, and NANOG in neuronal development and neuron differentiation. ZN462 performs an antioncogene function through binding to the DNA of oncogene PBX1 and preventing its heterodimerization and activities [72]. *DEK* encodes a proto-oncogene protein, which takes part in the regulation of multiple cellular processes. It is crucial for self-renewal, maintaining homeostasis of hematopoietic stem and progenitor cells, proliferation, differentiation, senescence, and apoptosis. The most important function of DEK is the promotion of DNA lesion repair and protecting cells from genotoxic agents, which typically induce unacceptable poly(ADP-ribose) polymerase (PARP) activation [73]. DEK interacts with the *NUP214* gene and forms the DEK-NUP214 fusion gene complex, which contributes to cellular transformation. At the same time, nonfusion DEK protein supports stem cell maintenance [74]. *PAWR* (Pro-Apoptotic WT1 Regulator) encodes a tumor suppressor protein that selectively induces apoptosis in cancer cells. GO annotations that are related to PAWR include the enzyme-binding and transcription corepressor activity. PAWR induces the switching from the autophagia to apoptosis selectively in cancer cells by down-regulating the antiapoptotic protein BCL2. These interactions sensitize cells to apoptotic stimuli and cause the regression of tumors [75]. The *MAL* gene encodes Myelin and lymphocyte protein, a membrane protein from the MAL family of proteolipids, localized to ER in neurons. MAL participates in the apical transport of proteins in polarized epithelial cells [76,77]. Gain- and loss-of-function assays showed that MAL is a tumor suppressor; it inhibits the proliferation, invasion, and EMT in cancer cells. EMT has been confirmed to occur in many cancer types. In tumor cells, it leads to the loss of the epithelial phenotype and acquisition of mesenchymal characteristics [78,79]. However, *MAL* knockdown promoted EMT-like changes [80]. *p53* encodes a phosphoprotein, which along with ATM and Gadd45a take part in the DNA damage cellular response pathway and maintain genomic integrity. p53 is known as the tumor suppressor protein that prevents the outgrowth of aberrant cells by inducing cell cycle arrest, DNA repair, or programmed death and regulates the progression of the cell cycle and cell differentiation. The decrease in p53 can impact significantly on how ARPE19 cells deal with damage. The reduced levels of p53 on several endpoints, including cell survival, as expected, can lead to a decrease in p21/waf1 transcription in proliferating and differentiating cells [81]. *ILF2* encodes protein, which induces the weakening and loss of cell adhesion to the ECM. The main biological function of ILF2 is the control of the cell cycle and apoptosis [82,83]. PTEN governs several cellular processes, including survival, proliferation, energy metabolism, and cellular architecture, by suppressing the phosphoinositide 3-kinase (PI3K)–AKT–mammalian target of the rapamycin (mTOR) pathway through its lipid phosphatase activity [84]. *PCYT1A* encodes Phosphate Cytidylyltransferase 1A Choline, which is involved in the regulation of phosphatidylcholine biosynthesis. De novo phosphatidylcholine synthesis is required for autophagosome membrane formation and maintenance autophagy [85]. PCYT1A suppresses cell proliferation and migration by inhibiting the mTORC1 pathway in some cancers [86]. It is interesting that the mTORC1 pathway is a key upstream signal that mediates tissue and organ regeneration and translational control. This pathway is in a highly labile state that is specific to urodele amphibians. These mechanisms ensure the rapid activation of the mTORC1 pathway compared with mammals in response to injury [49]. The *Smurf1* gene encodes E3 ubiquitin ligase, which is involved in fibrosis control via ubiquitination and proteasomal degradation proteins and regulates a variety of cellular activities, especially in the nervous system and innate immune responses. Previous studies have demonstrated that protein ubiquitination is an important regulatory post-translational modification controlling (OS, fibrosis, and EMT. Smurf1 is a negative feedback regulator for IFN signaling by targeting STAT1 for ubiquitination and proteasomal degradation, which attenuates the immune responses [87]. Smurf1 facilitated OS and fibrosis by promoting Nrf2 ubiquitination and degradation [88,89,90]. It was proven to have an essential role of PTEN in multiple biological processes, including the regulation of genomic instability, DNA repair, cellular senescence, and cell migration, besides being a well-characterized tumor suppressor [91]. A previous report has shown that down-regulation of the expression of *PTEN* in ARPE-19 cells under stress conditions disrupted intercellular adhesion, triggered DNA damage, and increased the apoptosis by activating the p53-dependent pathway. However, an overexpression of *PTEN* increased cell survival by suppressing p-H2A in response to DNA damage and apoptosis [92].

### 3.5. Metabolic- and Oxidative Stress-Related Cellular Processes

Retinal detachment as a result of injury is accompanied by the expression of high levels of ROS and/or heat shock proteins and, as a consequence, the expression of the related genes. This reprogramming is an energy-consuming process that involves de novo synthesis of nucleotides and depends on the level of ROS and calcium signals. During the process of cell dedifferentiation, the changes occur in the system of second messengers. These processes affect the balance of intracellular and extracellular nucleotide levels [93,94]. The reprogramming of the RPE cells in newts includes an activation of stress-responsive genes, which are components of endogenous cell protection systems. In our study, in the RPE at the early stage of reprogramming (4 days), the up-regulated transcription for the gene *IDH2*, which encodes NADP(+)-dependent isocitrate dehydrogenase, was detected. RNA-sequencing analysis demonstrated that the key antioxidant transcription factor Nrf2 specifically up-regulated the transcription of isocitrate dehydrogenase 2 (IDH2), a key enzyme regulating tricarboxylic acid cycle and mitochondrial function [95,96]. According to transcriptome analysis, the early stages of RPE reprogramming are accompanied by an increase in the transcriptional activity of *AT1B1* (AT1B1 family of Na^+^/K^+^ and H^+^/K^+^ ATPases beta chain proteins) and *ATP5H* (ATP synthase subunit d, Mitochondrial). ATP5H catalyzes the synthesis of ATP from ADP using an electrochemical proton gradient across the inner membrane during oxidative phosphorylation [97]. The de novo synthesis of nucleotides is strictly regulated at multiple levels, such as the regulation of precursors and of enzymes at the transcriptional and post-translational level, under the control of ADP-ribose pyrophosphatase (NUDT9) and transcription factors, including C-MYC and FOXK2 [98,99,100,101,102,103,104]. Ca^2+^ is one of the earliest wound-induced signals, which can induce ROS signaling in multiple species [105,106]. It was shown in a previous study using a model of conversion of pigmented iris cells that intracellular [Ca^2+^] accumulation is mediated by cAMP during cell dedifferentiation [107]. The mitochondrial Calcium Uniporter MCU-1 is essential for rapid mitochondrial Ca^2+^ uptake and mtROS production after injury. mtROS promotes cytoskeletal remodeling in wound healing and wound closure by locally using the mechanisms of inhibition of Rho GTPase activity via a redox-sensitive motif [108]. OS is one of the mechanisms of cells’ response to damage, and it occurs in cells undergoing an apoptosis, even if the proliferation and cell reprogramming are subsequently initiated [109,110]. ROS are supposed in protecting cells during the wound healing when their concentration is within the physiological norm [111,112]. It was shown that ROS (H_2_O_2_, hydrogen peroxide) production is rapidly induced after wounding [113,114] and is required for tissue regeneration in several species (zebrafish, frogs, salamanders, etc.) [115,116,117]. Brain injury in adult newts triggered reparative neurogenesis that is required for the compensation of dying neurons. But the intensity of this process decreased when ROS production was inhibited [118]. The data from the above-cited works demonstrate that animal species with a high regenerative potential can use ROS to activate tissue repair programs.

### 3.6. Proteins with a Chaperone Function

The reprogramming of the RPE activates the transcription of a number of genes encoding proteins that perform cell protection and chaperone functions in response to cellular stress. According to our transcriptome data, an increase in the transcripts corresponding to genes encoding *CYB* (Mitochondrially Encoded Cytochrome B), *FRIHB heavy subunit of ferritin*, *GLNA*, *SMURF1*, *SEPP1*, *SRPRB*, *FOXK2*, *INO1*, etc., in RPE samples at 4 days of RPE reprogramming were detected. Several of these genes encode a wide range of stress-related proteins, which can act as the chaperones EDEM3, ENPL, CCT5, FRIHB heavy subunit of ferritin, INO1B, SEPP1, and NSA2, or can act as the co-chaperones BAG6, MDN1, DNJA1, etc. Thus, the overexpression of *EDEM3* encoding the ER degradation-enhancing Alpha-Mannosidase Like Protein 3 has a cyto-protective effect against ER stressors. And on the contrary, the depletion of *EDEM3* in cancer cells induces an ER stress transcriptomic signature [119]. *FTH1* encodes the ferritin-heavy chain (FTH1), which is the component of the molecular machinery that protects tissues against the toxic products of spontaneous oxidation of free Fe(II), insoluble Fe(III), and ROS. In addition, FTH1 exerts significant antigrowth effects in breast cancer cells by inhibiting the expression of c-MYC. The activation of *GLNA* (glnA), which encodes a key enzyme, glutamine synthetase, is associated with the regulation of nitrogen metabolism [120]. SRPRB (SRP Receptor Subunit Beta) is a transmembrane protein, which includes GTP binding and GTPase activity. It takes part in pathways related to the unfolded protein response (UPR) and cellular responses to stimuli [121]. The activation of *SEPP1* is involved in the antioxidant-mediated protection and plays a crucial antitumor role [122]. The increase in INO1B was revealed in RPE at 4 days after retinal detachment. It was previous demonstrated that the unfolded protein response (UPR) induced an activation of *INO1B*, which maintained the redox balance in the cell, functioned as a redox-switch, and protected cells from OS [123]. According to transcriptome data, an increase in the level of transcripts for CAPR1, PPM1G protein phosphatase, Mg^2+^/Mn^2+^ dependent 1G, GSTA3, etc., was revealed in RPE at 7 days of reprogramming. *CAPR1* encodes GAP1 mRNA-binding protein caprin-1, a member of the conserved proteins from the family of GTPase-activating proteins (GAPs, cell cycle-associated protein 1), which activate in response to stress. GAPR1 acts as a regulator of mRNA transport, translation and/or stability, and the synaptic plasticity in neurons. GAPR1 performs the function of the calcium-binding chaperone and promotes folding, assembly, and quality control in ER via the calreticulin/calnexin cycle [124]. CAPR1 can bind selectively with a subset of mRNAs, for example MYC and CCND2 mRNAs, in the control of the proliferation and cell migration [125,126]. *PPM1G* encodes the protein phosphatase Mg^2+^/Mn^2+^ dependent 1G, a member of the PP2C family of Ser/Thr protein phosphatases, which are known to be negative regulators of cell stress response pathways. PPM1G is found to be responsible for the dephosphorylation of pre-mRNA splicing factors, which take part in the formation of functional spliceosome [127] and contribute to cell proliferation and tumor transformation [128].

We have revealed a high level of transcription for *HSP7C* and *HSP90B1* in RPE at 7 days of reprogramming. *HSP7C* (HSPA8, Heat Shock Protein Family A (Hsp70) Member 8) is a protein coding gene. This molecular chaperone plays a pivotal role in the protein quality control system, ensuring the correct folding of proteins and the refolding of misfolded proteins and controlling the targeting of proteins for subsequent degradation. HSP7C has been shown to be involved in a wide variety of cellular processes (protection from stress, folding and transport of newly synthesized polypeptides, chaperone-mediated autophagy, activation of proteolysis of misfolded proteins, formation of protein complexes). This is achieved through the cycles of ATP binding, ATP hydrolysis, and ADP release, mediated by co-chaperones [129]. ATP-independent and ATP-dependent chaperone HSP protect the cells from cellular stress (OS, ER stress, ischemia [130]). Previously, the participation of HSP70 and HSP90 (at the mRNA and/or protein level) in the retinal regeneration in *P. waltl* has been shown [131]. The increase in the transcriptional level of *GSTA3* (Glutathione S-transferase A3) and AGR2 was revealed at 7 days of RPE reprogramming. Glutathione S-transferase A3 is known as an antioxidative protease. GSTA3 is considered to exhibit hydroperoxidase activity in vivo, which decreases the excess ROS and free-radical-induced lipid peroxidation (LPO) accumulation [132]. GSTA3 inhibited hepatic stellate cells (HSCs) activation and liver fibrosis through suppression of the MAPK and GSK-3β signaling pathways by regulating OS [133]. The TCPE molecular chaperone, containing the TCP1 complex (CCT), which is encoded by the *CCT5* gene, is among the molecular chaperones, and is activated at 7 days of RPE reprogramming. It is a member of the chaperonin family [134]. Unfolded/misfolded proteins arising in the ER induces the activation of ENPL ER chaperones and endoplasmines [135] under the influence of ER stress and hypoxia [136,137]. *NSA2* encodes the nucleolar protein NSA2 Ribosome Biogenesis Factor (Nop seven-associated 2, also known as TINP1 (TGF-β inducible nuclear protein 1)), a secreted chaperone that increases the transcriptional level in RPE cells at 7 days of reprogramming. The obtained data demonstrated that NSA2 performs the function of the proliferation regulator, being the cell cycle repressor, and activating under stress conditions [138]. An increase in the transcript number for the *AGR2* gene was detected. *AGR2* encodes a protein, which is a member of the disulfide isomerase (PDI) family of ER proteins, which catalyzes protein-folding and thiol-disulfide interchange reactions. AGR2 performs the function of the molecular chaperone, which regulates the folding, trafficking, and assembly of cysteine-rich transmembrane receptors. It was also implicated in inflammation and cancer progression and in the regulation of the cell migration and transformation [139] by suppressing tumor inhibitor p53 [140]. The *NACA* gene was shown in the list of down-regulated genes at 7 days of RPE reprogramming. *NACA* encodes a protein that is associated with basic transcription factor 3 (BTF3) to form the nascent polypeptide-associated complex (NAC). NAC can bind to nascent proteins, which lack a signal peptide motif as they emerge from the ribosome. This mechanism blocks the interaction with the signal recognition particle (SRP) and prevents their mis-translocation to ER [141].

### 3.7. Chromatin Remodeling

The dynamic and strictly coordinated regulation of chromatin organization is essential for spatiotemporal and appropriately coordinated gene expression during development and regeneration [142]. The transcriptome analysis revealed the transcriptional activation of a number of genes that are involved in the remodeling of chromatin activity (*MIER1*, *CAPR1*, *SET*, *CAPR1*, *ZN462*, *SMARCA2* et al.). These genes encode the multifunctional proteins that are involved in the early cell response to injury and can also perform histone chaperone functions. MIER1 is the transcriptional repressor, which ensures chromatin silencing. It regulates the expression of a number of target genes, probably through the recruitment of a histone deacetylase HDAC1 and the highly conserved canonical ELM2-SANT domain [143]. *MIER1* down-regulation, in turn, promotes cell cycle gene expression through chromatin remodeling and supports proliferation and regeneration [144,145]. *SET* encodes the SET binding protein 1 Nuclear Proto-Oncogene (SETBP1). This multifunctional protein is involved in regulation transcription, protein phosphatase activity, nucleosome assembly and histone binding, and apoptosis and acts as a chaperon [146,147]. *ZN462* (Zinc Finger Protein 462) is a nuclear factor that encodes the protein that belongs to the C2H2-type zinc finger family. It contains multiple C2H2-type zinc fingers and is involved in transcription by regulating chromatin structure and organization [UniProtKB/Swiss-Prot Summary for ZNF462 Gene; 72]. *SMCA2* encodes the protein SMARCA2 (SWI/SNF Related, matrix-associated, Actin-dependent regulator of Chromatin, Subfamily A, Member 2), which is a part of the large ATP-dependent chromatin remodeling complex SNF/SWI (BAF chromatin remodeling complexes, including the family of BAF proteins). The switching of SNF/SWI activities is required for the transcriptional activation of genes, which is normally repressed by chromatin. The members of this family, besides ATPase activity, also exhibit helicase activity and are thought to regulate the target genes’ transcription via chromatin remodeling in an ATP-dependent manner. SWI/SNF contributes to the regulation of the neural progenitors-specific chromatin remodeling complex (neural progenitor BAF complex ACTL6A/BAF53A and PHF10/BAF45A) and the neuron-specific chromatin remodeling complex (neurons BAF complex ACTL6B/BAF53B and DPF1/BAF45B or DPF3/BAF45C). The signaling pathways that are related to SMCA2 include the regulation of gene transcription and chromatin organization. SMCA2 is hypothesized as the global transcription activator that is essential for the regulation self-renewal/proliferative capacity of the multipotent neural stem cells and their exit from the cell cycle. These mechanisms ensure a switch from a stem/progenitor to postmitotic cells as neurons exit the cell cycle, become committed, and acquire their differentiated state. ATP-dependent chromatin remodeling complexes (CRCs) and chromatin organizers (e.g., CTCF) demonstrate highly conserved mechanisms [148,149,150,151]. The identification of the regeneration-dependent regulatory elements (RREs) in the genome also supports this idea [152]. The stage-specific *RREs* activation and their epigenetic modification argues that certain regeneration-responsive loci in the genome can be subjected to heritable changes in chromatin organization [153] and can also be taxon-specific. Up-regulated genes/proteins significantly enrich signaling cascades that are associated with the formation of conditions for protein processing in ER, folding and muscle contraction, regulation of spliceosome and ribosome assembly, initiation of translation and apoptosis, regulation of transposon activity, cellular respiration and response to stress, and immune response (Appendix A).

### 3.8. Ribosome Biogenesis, RNA Translation, and Protein Processing

Among the down-regulated genes/proteins, there is a significant number of ribosomal proteins and proteins that regulate ribosome assembly, which in the Reactom DataBase are expressed in a large number of signaling cascades that include ribosome assembly as an integral part of a broader process. In all cases where these processes showed enrichment predominantly or exclusively at the expense of ribosomal proteins, we assessed them as signaling pathways of rRNA processing and translation. In other cases, a significant enrichment by down-regulated genes can be noted, i.e., decreased activity of the termination of the serine/threonine (O)-linked glycans biosynthesis pathways (involving the GalNAc-peptide linkage), TNF Signaling, C-type Lectin Receptors signaling, and VEGFA VEGFR2 Signaling (Appendix A). The analysis of the significant enrichment of the corresponding functional classes demonstrated inhibited translation of the enrichment mRNAs that encode the ROS-defense proteins, which are proteins that are involved in ribosome biogenesis and rRNA processing during RPE reprogramming. Among the genes involved in growth-related functions that are enriched in the translationally down-regulated group were the ribosomal genes and genes encoding proteins that are involved in the translation and regulation of protein synthesis (Figure 7 and Figure 8). The ribosomal proteins are the main molecular components in the control of rRNA processing in the nucleus and cytosol in ribosome biogenesis and cellular metabolism. RPL5 binds non-ribosome-associated cytoplasmic 5S rRNA to form a stable complex called the 5S ribonucleoprotein particle (RNP). RPL5 is necessary for the transport of 5S rRNA non-ribosome-associated cytoplasmic 5S rRNA to the nucleolus for assembly into ribosomes [154], and it may also function to inhibit tumorigenesis through the activation of downstream tumor suppressors and the down-regulation of the oncoprotein expression [155]. The negative co-regulation of the transcription and the translational machinery genes during the stress response has also been demonstrated [156]. The regulation of translation by redox switches of ribosomal proteins is a relevant molecular mechanism for cells’ viability and activates the multiple stress response signaling pathways [157]. Thus, the translation attenuation in response to mitochondrion-derived OS appears to be preserved in higher eukaryotes. The decrease in translation has been shown to occur upon an increase in endogenous ROS levels, which was caused by mitochondrial dysfunction and is likely to occur during the elongation and/or termination phase of protein synthesis [158]. The attenuation of stress-induced global translation favors the selective synthesis of proteins that aid in the recovery from stress. It is important that the selective translation is an integral part of mTOR signaling, a key pathway for maintaining the regeneration [159].

The growth in the proportion of down-regulated ribosomal and translation-associated proteins in our study suggests that a certain threshold of lower cytosolic translation is needed to activate the recovery in response to stress. The increase in the number of down-regulated ribosomal proteins as part of the ubiquitin ligase inhibitor activity group is consistent with our general view of the regeneration process development. These proteins regulate the signal transduction by the p53 class mediator and mesenchymal stem cell migration through interaction with partner proteins in signaling cascades. They inhibit the formation of the mitotic anaphase-promoting complex and metaphase/anaphase transition. p53 (TAp73) appeared to be important in maintaining the translation of mitochondrial transcripts, ribosomal biogenesis, and protein synthesis under conditions of elevated ROS levels, thus suggesting its significance for homeostasis [160]. The biological meaning of the expression genes’ activity, which determines the translation efficiency at the same time as a simultaneous decrease in the activity of genes encoding proteins that are the structural components of ribosomes, remains unclear. This cell reaction may be a compensatory response of the genome, meant to decrease the translation efficiency, which is caused by a decrease in the number of newly assembled ribosomes. The second explanation could be the formation of a specific filter that ensures that an increase in the selectivity of the access new activated ribonucleoprotein complexes to the ribosome under conditions of increased competition. In fact, down-regulation of the ribosomal and associated translation proteins can contribute to the reproduction and regulation of developmental processes and retinal regeneration.

Rapid protein biosynthesis during RPE cells reprogramming can be accompanied by the production of damaged proteins that must be removed by protein quality-control mechanisms. An overall reduction in translation processes under the cells’ reprogramming may decrease the load of damaged proteins. In addition, translation is one of the most energy-consuming cellular processes [161], and the decreased translation intensity at the time points can mobilize energy for cellular viability maintenance and repair processes [162]. In response to cellular stress, the translation-modulating pathways form a network that coordinates transcriptional and post-transcriptional mechanisms for overcoming pathological conditions that can compromise protein homeostasis. It is assumed that the cooperative action of the described molecular processes can cause the weakening or blocking of signals that stabilize RPE cell differentiation and switch to a program of proliferation and production of retinal neuroblasts.

The determination of the transcription factors that are involved in the regulation of the activity of evolutionary orthologs of the newt genes was not possible due to a high variability in enhancers, which only have a 1% similarity within the mammalian class. Enhancer similarity in more evolutionarily distant groups is an extremely rare phenomenon [163]. However, the signaling cascades, which determine the biological processes, are evolutionarily conservative.

The successful advancement of RPE reprogramming is ensured by changes in cel-lular metabolism that are aimed at implementing the regulation of glycolysis, as well as biosynthesis of nucleotides, proteins, and lipids. The elucidation of the features of the reprogramming process in RPE cells at the transcriptome level may become an important “touch” to characterize the “portrait” of this phenomenon in newt. Reprogramming and metabolic rearrangement in RPE cells are quite complex, and the detailed mechanisms underlying them remain to be clarified. It is likely that certain features of endogenous defense systems and cellular metabolism in Urodela make RPE cells resistant to stress, allowing them to overcome the negative influence of OS, or, conversely, to use it as a significant part of the mechanism, ensuring a successful RPE reprogramming [164,165,166]. The data we have obtained at the bioinformatics level support these ideas, which need further experimental confirmation. Data on transcriptome analysis demonstrate that most of the molecular components of the transcriptome signature are involved in multiple metabolic and signaling pathways and protein complexes. The implementation of multiple functions requires the tight regulation of factors in the tissue-specific microenvironment and organism’s regulatory systems.

## 4. Materials and Methods

### 4.1. Newt Strain and Surgical Operations

The experiments were carried out on Urodela (*Pleurodeles waltl*), who were bred in the aquarian IBR RAS. This study used adult newts (9 months). All experiments were carried out in accordance with the guidelines approved by the IBR RAS Animal Use and Care Committee on Bioethics of the IBR RAS. N.K. Koltsov. Surgical detachment of the NR from RPE was performed as described previously [30,167]. Operation and sacrifice were carried out under anesthesia (anesthetic: tricaine methanesulfonate MS 222 (Sigma-Aldrich, St. Louis, MO, USA) in 0.65% saline solution (NaCl 1:1000)). Mechanical detachment of the retina from RPE was performed according to the standard previously developed method. In laboratory, the animals had been kept in special containers/aquarium tanks at 20 °C under a natural light condition. During anesthesia, the animals were placed in the dark at room temperature (20–22 °C) for 2 h, allowing for darkness adaptation of the retina that makes the adherence between the RPE and NR the weaker. After this treatment, the animals did not demonstrate a pupillary reflex during surgery and were not awake for at least 4 h. No other in vivo experiments were carried out. The stage of retinal regeneration and corresponding post-operative (po) day were determined according to previous criteria [30].

### 4.2. Preparation of Tissue Samples

To obtain the de novo assembly transcriptome involved in early processes of retinal regeneration, including, first of all, RPE reprogramming, as well as cell cycle re-entry/proliferation, the samples of RPE at early stages after retinal detachment (time po: 0 days; 4 days; 7 days) were analyzed. The scheme of experiment is illustrated in Figure 1. The samples from eye tissues were prepared as described previously after the newts were sacrificed under anesthesia to minimize suffering [30]. To collect RPE, the animals were decapitated, and the eyeballs were carefully enucleated with special microsurgery instruments for small animals. The eyeballs were first dissected into the eye cups (posterior half of the eyeball), which included the complex RPE-choroid-sclera. The RPE was isolated from the collected samples using a method that leads to a high yield of RPE-derived RNA while preserving its quality [168]. The sample RRE from the right eye were used for the control sample (native), and the left eyeballs were used for the 4 days po (early stage of transdifferentiation) or 7 days po (late stage of transdifferentiation) samples. RPE samples were collected in IntactRNA for RNA stabilization and protection from degradation (Evrogen, Moscow, Russia).

### 4.3. RNA Extraction

Gene profile changes were assessed in comparison with native RPE. We isolated RPE cells (native) and RPE-derived cells (4 days, early; 7 days, later) from intact eyeballs and from eyeballs after retinal detachment, respectively. Tissues of the 8 regenerates at each time point were pooled and RNA was extracted from regenerating tissues and control tissue. Reagent “Extract RNA”, which utilizes guanidine thiocyanate (Evrogen, Moscow, Russia), was used to isolate RNA from RPE tissue, and then, RNA was treated with DNase Turbo (Thermo Fisher Scientific, Waltham, MA, USA). Total RNA from RPE tissue was extracted using the ”Extract RNA”, which utilizes guanidine thiocyanate (Evrogen, Russia), and DNA was removed with TURBO DNA-free (Thermo Fisher Scientific, Waltham, MA, USA). This made it possible to ensure the most complete extraction of RNA from tissue and obtain undamaged, highly purified RNA, without contamination by genomic DNA impurities. RNA concentration was assessed using NanoDrop spectrophotometer 2000 NanoDrop spectrophotometer 2000 (Thermo Fisher Scientific, Wilmington, DE, USA). The concentration of the RNA was determined using a NanoDrop spectrophotometer 2000 (Thermo Fisher Scientific, Wilmington, DE, USA). An Agilent 2100 Bioanalyzer was used to control the quality of the extracted total RNA sample (RIN value greater than 7) (Agilent Genomics, Santa Clara, CA, USA). RNA quality was measured with NanoDrop spectrophotometer 2000 (Thermo Fisher Scientific, Wilmington, DE, USA). The OD260/280 ratio was about 2.0. The presence of DNA was not detected in the samples. Next, the RNA was used to prepare cDNA libraries for NGS sequencing of the RPE transcriptome.

### 4.4. cDNA Library Construction

Samples of RPE cells for analysis were obtained. RNA-seq libraries for transcriptome sequencing were prepared using the MGI Easy RNA Library Prep Set (MGI Tech Co., Ltd. Shenzhen, China). The kit is optimized for converting 10 ng^−1^ μg eukaryotic total RNA into a single-stranded circular DNA library for gene expression profiling and transcriptome analysis. For sequencing, 400 ng of RNA was taken from each sample. RNA quality and integrity were assessed at each step using an Agilent 2100 Bioanalyzer (RIN value greater than seven) (Agilent Genomics, Santa Clara, CA, USA). DNA library preparation procedures were performed using the MGI Easy RNA Library Prep Set (MGI Tech Co., Ltd.). The principle of library preparation is based on the circularization of DNA fragments using ligated adapters and further amplification of DNA fragments of the sample, like a rolling ring. The synthesized long strand of DNA is folded into a compact spherical nanoball structure (“DNA-NanoBall”) after the number of copies of the original template reaches more than 300 copies. The approach used allows us to avoid the accumulation of DNA polymerase errors during amplification of target fragments. Next, the MGI Easy rRNA Depletion Kit (MGI Tech Co., Ltd.) was used to remove ribosomal RNA and enrich the mRNA. The quality and integrity of cDNA was checked using a Nano Drop 2000 spectrophotometer and via electrophoretic separation of fragments. cDNA quality was checked at each step on an Agilent 2100 Bioanalyzer (Agilent Genomics, Santa Clara, CA, USA).

### 4.5. Sequencing

This study involved 6 samples RPE (2, 3, 4, 5, 6, 7), with two replicates for each analyzed point. Each sample included total mRNA from 8 tissues. RNA sequencing was performed on the DNBSEQ-G400 platform (MGI Tech) to generate 300–400 million single-end reads, with 250 basepair reads per sample.

### 4.6. Transcriptome Assembly and Annotation

De novo transcriptome assembly (obtaining sequences of sequenced transcripts) preceded the assessment of the molecular profile of the RPE and was carried out using the Trinity package program (https://github.com/trinityrnaseq/trinityrnaseq/wiki, accessed on 5 October 2023) (Appendix A). We used the transcriptome assembly methodology for RPE *P. waltl*, which is generally accepted in comparative genomics for animals that are considered “non-model species” with unstudied or insufficiently studied genomes or transcriptomes (“genome-guided assembly” or “alignment-based assembly”). Reads of 250 nucleotides in length were mapped to known homologous genes, including their various splice variants, and further banded into a whole transcript. Blast2GO was used for GO analysis and the functional annotation of contigs. TransDecoder and Trinotate were applied to identify and annotate ORFs.

### 4.7. Data Analysis

The search for genes that claim to be regulators of RPE cells transdifferentiation in the newt is required for the multistep evaluation and filtering, reducing the number of sequences without losing those that would be of biological significance. In the context of de novo transcriptome analysis, after filtering the encoded sequences from STRs (short tandem repeats), we searched for proteins that were orthologous to the analyzed sequences, which allows us to judge their functions more accurately. The annotation was carried out using several databases of closely related proteins for amphibians as well as evolutionarily distant species (references). As a result, information was obtained about cellular processes in which orthologous proteins take part, which allows us to put forward a hypothesis about the functions of the analyzed proteins. The National Center for Biotechnology Information’s (NCBI) GEO database (https://www.ncbi.nlm.nih.gov/geo/, accessed on 6 September 2023) was used. We also used open access databases on amphibians [24,105,169]. All gene homologues were found using annotated sequences, and they were strictly corresponded. To predict coding regions (potential peptides) in transcripts, the Transdecoder program was used (http://cole-trapnell-lab.github.io/cufflinks/announcements/transdecoder/, accessed on 6 September 2023). The founded open reading frames (ORFs) of nucleotide sequences that could potentially encode proteins were compared with homology to known proteins using the blastp and hmmscan programs. The eukaryotic part of the Uniprot database (https://www.uniprot.org/, accessed on 2 September 2023) (Appendix A) was used as a protein database. It should be emphasized that a mass of almost identical transcripts (isoforms + artifactual variants) is generated, which complicates many downstream analyses, when assembling deeply sequenced libraries. Peptide clustering was carried out in the cd-hit program with a similarity threshold of 90% to reduce degeneracy (Appendix A). An approximate assessment of the protein composition was carried out using the BUSCO v5.4.3 database (Appendix A).

### 4.8. Differential Gene Expression Analysis

Differential gene expression (DEG) was assessed in samples of 4-day (“early”) and 7-day (“late”) stages of RPE cell reprogramming and compared with intact RPE tissue served as a control, taking into account data on the number of reads per sequence (Figure 2). DEGs were detected using the Trinity (RSEM) (14.0.11 release) and edgeR (v0.1.0 release) software packages (Appendix A). The genes that met the cutoff criteria (*p*-value < 0.05 and |log2 fold-change (logFC)| > 2.0) were designated as DEGs. Heatmap was used to visualize the differential gene expression profile between the three studied groups. To evaluate a heatmap of DEGs, the edgeR package was used (transcripts are indicated in the rows, samples are indicated in the columns) (Figure 2). Venn diagrams were applied to detect the intersecting parts of the DEGs between “norm vs. early” and between “norm vs. late”. Shared DEGs between compared and unique transcripts were used in downstream analysis. MA plot was used to visualize DEGs for all groups. It demonstrates the dependence of the expression magnitude of genes/proteins on the degree of expressed change in compared samples (Figure 3).

### 4.9. Assessment of Nonrandom Enrichment of Signaling and Metabolic Cascades and GO-Biological Process

The biological process (BP), cellular component (CC), and molecular function (MF) of DEGs were assessed using GO enrichment analysis. Analyses were conducted using the GO knowledgebase (released 9 October 2023) and PANTHER Classification System tools. The significance of enrichment was confirmed at FDR values <0.05. All used libraries for this analysis were human-specific. We have used the human-specific libraries of Enrichr knowledgebase, REACTOME pathways, BioPlanet pathways, Wikipathways, Kyoto Encyclopedia of Genes and Genomes (KEGG) pathways, and Elsevier Pathways (Appendix A), to identify nonrandom enrichment of signaling and metabolic cascades with up- and down-regulated proteins. Statistical significance was assigned to terms with a *p*-value < 0.05. The adjusted p-value based on Bonferroni correction was employed to identify the statistical significance. Genes were clustered into groups according to biological processes in which they participate, as a result of GO annotation. To create and visualize a functionally grouped network of terms/pathways and their relations with the DEGs, the CytoScape program [170] and the ClueGO plugin [171] have been used. RNA-seq raw data can be downloaded from the NCBI SRA database (BioProject accession number: PRJNA1035927).

## 5. Conclusions

This is a pioneering transcriptome study of RPE cell reprogramming in *Pleurodeles waltl* adult newt retinal regeneration. The aim of the study was to explore the molecular mechanisms of RPE reprogramming that underlie newt retinal regeneration.

The performed analysis of the RPE RNA-seq transcriptome data covers only the first week of the retinal regeneration process after its experimental detachment, including the period of dedifferentiation and conversion of RPE cells. For the late stage of RPE reprogramming, an enrichment of overexpressed genes for processes that are characteristic of inflammation, cell adhesion, intercellular contacts, ECM reorganization, cell division, apoptosis, and differentiation were revealed, i.e., the range of basic characteristics that are inherent in regeneration processes. We have revealed that RPE reprogramming in *P. waltl* is accompanied by the activation of OS-related genes of the early response to injury, chaperones and co-chaperones, and genes involved in the rearrangement of the protein biosynthesis via the suppression of oncogenes and activation of suppressors of oncogenes and EMT. We supposed that these mechanisms might be important for successful RPE reprogramming followed by retinal regeneration in newt. The initiation of RPE reprogramming is focused on the restructuring of the cell’s biosynthetic machinery for the generation of stem-like progenitor cells (neuroblasts). The down-regulation of the ribosomal proteins that are associated with translation during RPE reprogramming has been revealed. We hypothesized that a time point decrease in the translation processes’ intensity in reprogramming RPE is necessary to control the production of damaged proteins and mobilize energy to maintain repair processes.

Our findings have shown that the identified molecular and biological processes of RPE reprogramming in newt in response to injury can be preceded by a genetic program of neuroblasts production by these cells. It is likely that during the time period examined in the present study, the molecular programs of neurons and glia differentiation that ensured the retinal regeneration are suppressed in RPE cells and will be activated at subsequent stages of regeneration.

It is likely that features of endogenous defense systems in Urodela make RPE cells resistant to stress, allowing them to overcome the negative influence of OS, or, conversely, to use it as a significant part of the mechanism that ensures a successful reprogramming. The study deepens our understanding of the mechanism of RPE reprogramming that ensures retina regeneration in the newt, by providing information about genes and pathways that participate in this process. Transcriptomic analysis findings point to potential directions for future research. In the near future, it will be highly desirable to carry out verification via real-time PCR, phylogenetic analysis, and functional studies on the role of specific genes. Future work will allow us to solve these problems, using in vivo experimental models of retinal detachment and also in vitro techniques.

## Figures and Tables

**Figure 1 ijms-24-16940-f001:**
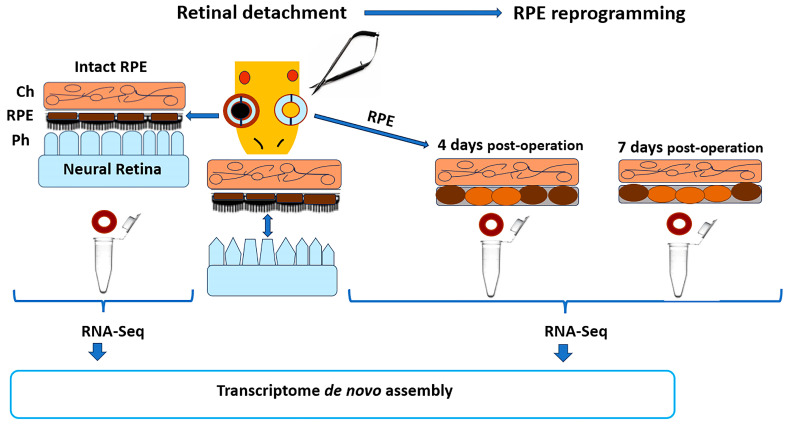
Strategy of the experiment for analysis of the RPE transcriptome after retinal detachment in adult newt *P. waltl*. RPE cells for transcriptome analysis are obtained at 4 and 7 days po (post-operation). RPE 4 days po, designated as an early stage of RPE cell reprogramming is characterized by the first subtle signs of depigmentation, indicating the lost their epithelial characteristics. RPE 7 days po, designated as a later stage of RPE cell reprogramming, is characterized as an increase in the number of partly depigmenting RPE cells that are still in the layer (see text for details). Ch—choroid; RPE—retinal pigment epithelium; Ph—photoreceptors.

**Figure 2 ijms-24-16940-f002:**
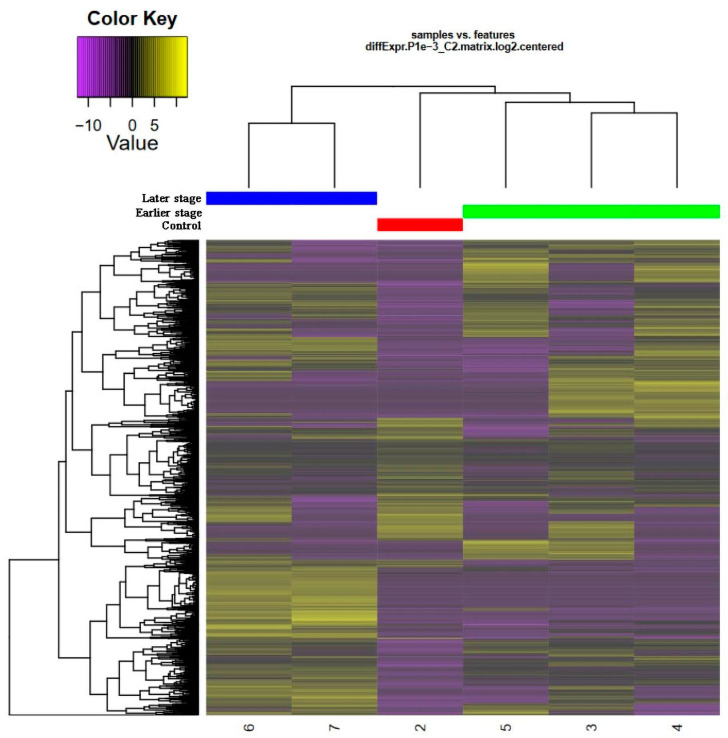
Heatmap of differentially expressed genes (DEGs)/proteins in the RPE during retinal regenerating in the newt *P. waltl* in 3 classes of analyzed samples, “norm” (red), “early” (green), “late” (blue). DEGs were assigned a color intensity with a 5-fold bound color key (log10 scale). Time point samples are color-coded and abbreviated as «late»; «early»; «norm».

**Figure 3 ijms-24-16940-f003:**
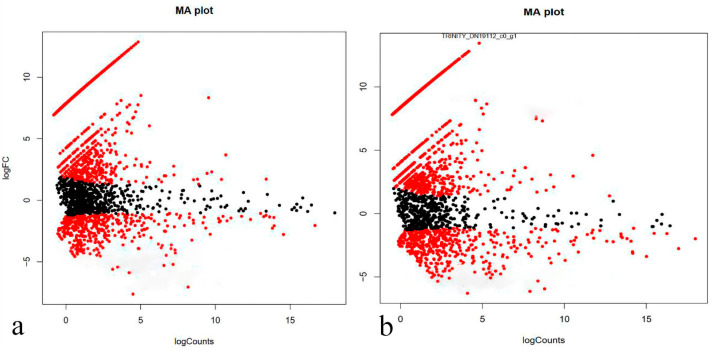
MA plot for differential expression analysis in samples of the early (**a**) and late (**b**) stages of RPE reprogramming relative to normal tissue. The X axis indicates the normalized mean, the Y axis indicates the log 2-fold change. Plot demonstrates the dependence of the expression magnitude of compared genes/proteins on the change expression degree. Genes that significantly changed their expression and decreased it (the lower part of the diagram) or increased it (the upper part of the diagram) were marked in red. Genes that did not change their expression were marked in black.

**Figure 4 ijms-24-16940-f004:**
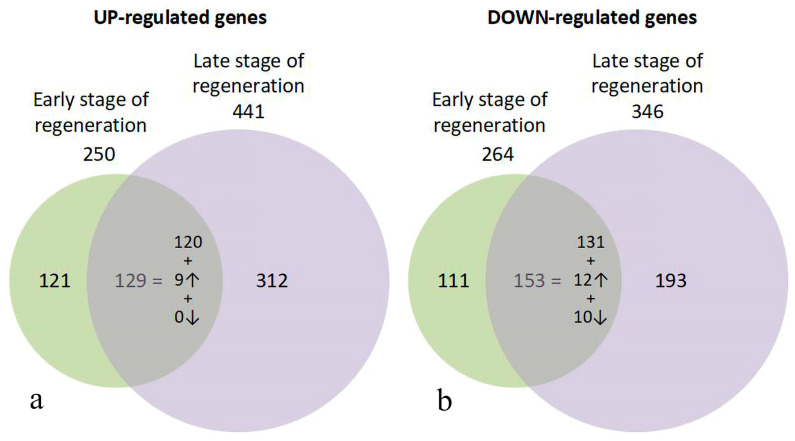
Venn diagram of the up- (**a**) and down-regulated (**b**) genes in early and late stage of RPE cell reprogramming during retinal regeneration in newt. The expressions in the regions of intersection of sets demonstrate the common DEGs for both stages of regeneration and the proportion of them with similar levels of differential expression, with an increased level at a later stage by two times or more or decreased by two times or more. The direction of the arrows indicates an increase or decrease in differential expression.

**Figure 5 ijms-24-16940-f005:**
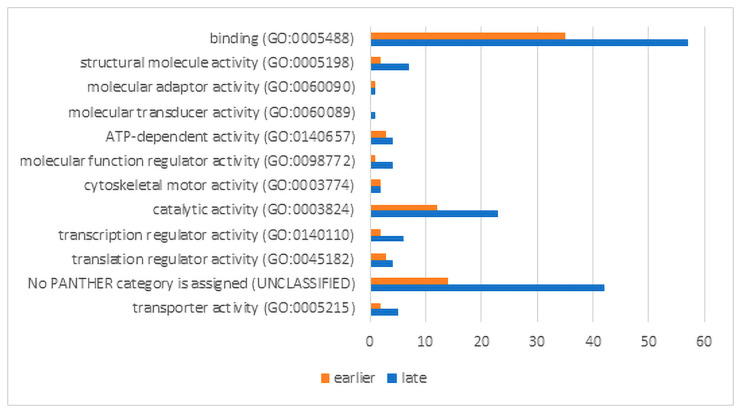
Molecular functions of up-regulated proteins at early and late stages of RPE reprogramming during retinal regeneration. The X axis shows numbers of gene hits.

**Figure 6 ijms-24-16940-f006:**
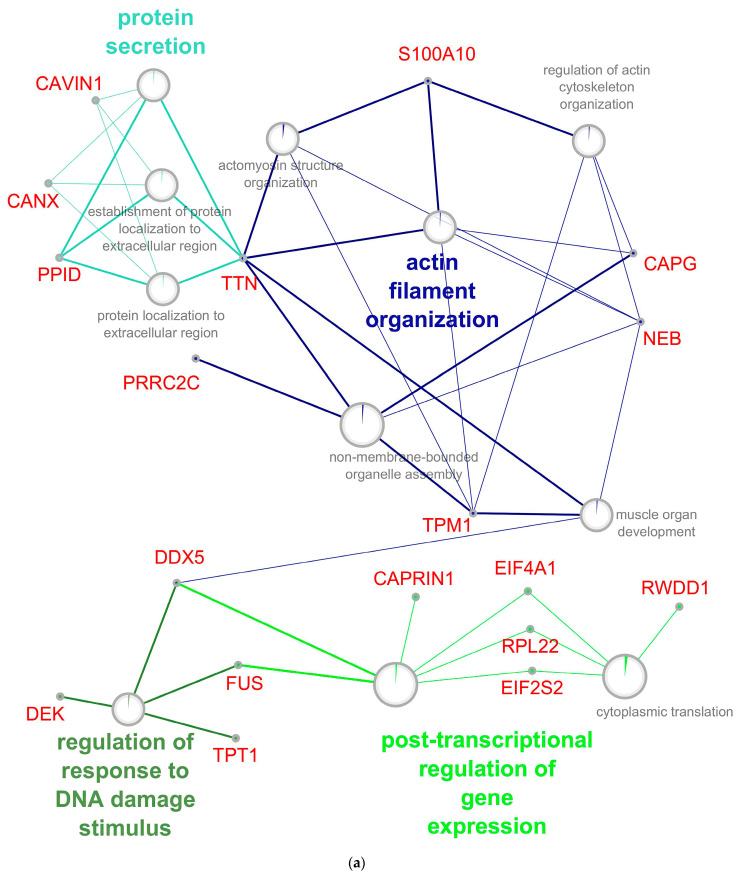
Network of up-regulated GO biological processes from early (**a**) and late (**b**) stage of RPE reprogramming during retinal regeneration. The networks were built using the CytoScape program. The processes, whose enrichment in up-regulated proteins/genes is confirmed by an FDR value of at least 0.05, are presented on the network. Sectors indicate the proportion of up-regulated proteins. Each process has its own color coding. The names of genes associated with the above processes are given in red font. The names of the clusters including related processes are given by the name of the process that has the largest absolute number of up-regulated proteins.

**Figure 7 ijms-24-16940-f007:**
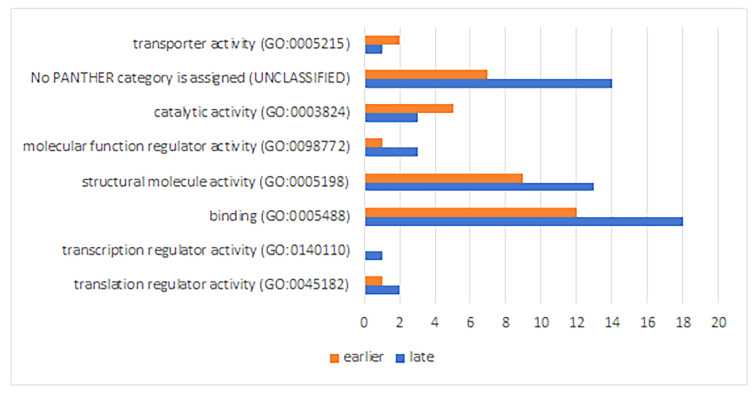
Molecular functions of down-regulated proteins at early and late stages of retinal regeneration. The X axis shows numbers of gene hits.

**Figure 8 ijms-24-16940-f008:**
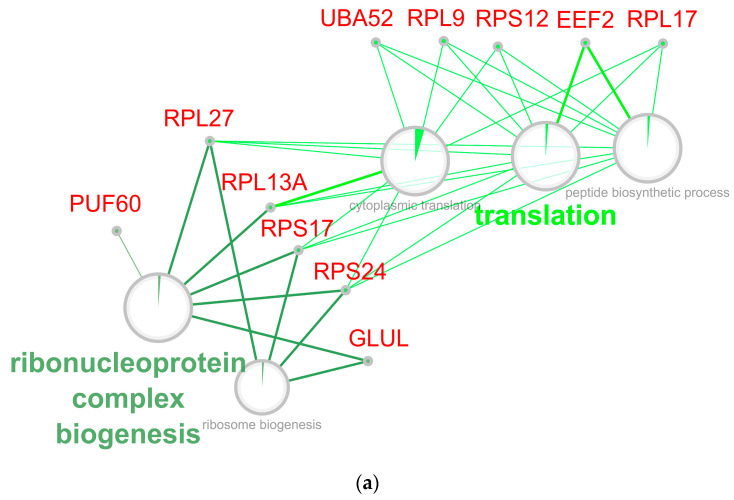
Network of down-regulated GO biological processes from early (**a**) and late (**b**) stage of retina regeneration. The networks were built using the CytoScape program. Processes whose enrichment in down-regulated proteins/genes are confirmed by an FDR value of at least 0.05 are presented on the network. Sectors indicate the proportion of down-regulated proteins. The names of genes associated with the above processes are given in red font. The names of the clusters, including the related processes, are given by the name of the process that has the largest absolute number of down-regulated proteins.

## Data Availability

The raw RNA sequencing data have been deposited in the NCBI Sequence Read Archive (SRA) under BioProject accession number PRJNA1035927.

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
