# Peer review of "Molecular Signatures Integral to Natural Reprogramming in the Pigment Epithelium Cells after Retinal Detachment in Pleurodeles waltl"

_ijms, 2023, doi:10.3390/ijms242316940_

Round 1
Reviewer 1 Report
Comments and Suggestions for Authors
This is a very good paper, with lots of results and a huge amount of carefully referenced data. It is well written and readable. The only thing I do perceive as a "problem" is that this is a sort of hybrid between a research paper and a review. In this sense, ten printed pages of Discussion alone can discourage a reader. I suggest to shorten it.
Minor points:
The retinal detachment is a traumatic event: is this accompanied by the expression of high levels of ROS and/or shock proteins? I would expect this and as a consequence the expression of the related genes.
I would expect also an increase of expression of Pol I-related genes. Is this correct?
line 502: The distribution and partial loss.. I think the authors should amplify the concept. To simply define it a "liberation" seems reductive. These are the true signals that consent the dedifferentiation.
line 168: regeneration and show. Remove AND
line 382: AN should be AND
Author Response
Dear reviewer. Foremost, we are very grateful for the review, valuable comments made and pointing out the omissions.
Comments and Suggestions for Authors
This is a very good paper, with lots of results and a huge amount of carefully referenced data. It is well written and readable. The only thing I do perceive as a "problem" is that this is a sort of hybrid between a research paper and a review. In this sense, ten printed pages of Discussion alone can discourage a reader. I suggest to shorten it.
- We have taken into account your recommendation. We have edited the text of manuscript, removing the redundant information in the section Discussion.
Minor points:
- The retinal detachment is a traumatic event: is this accompanied by the expression of high levels of ROS and/or shock proteins? I would expect this and as a consequence the expression of the related genes.
- As you rightly noted, the retinal detachment is accompanied by the expression of high levels of ROS and/or shock proteins. And indeed, our data support this fact. We have emphasized, that the retinal detachment as a result of injury is accompanied by the expression of high levels of ROS and/or heat shock proteins, and as a consequence the expression of the related genes.
We identified the up-regulation of gene RACK1A, which play a central role in the production of reactive oxygen species in response to damage [42]. This fact allows us to indirectly judge about the increase in ROS production at the early stages of the newt RPE reprogramming, after the retinal detachment.
- I would expect also an increase of expression of Pol I-related genes. Is this correct?
- Ribosome biogenesis commences with transcription of a variety of RNAs by RNA Polymerases (Pol I and III), and is accompanied by an increase of expression of Pol I-related genes. The approach we used did not allow us to identify the Pol I expression. This may be due to a decrease in ribosomal and translational activity. But we have identified upregulation of CTR9 Paf1/RNA Polymerase II Complex Component, which was associated with Pol II throughout the transcription cycle and also the related genes. Amoun them are the negative regulators of transcription PHF12 (PF1), translation of mRNAs and others.
- line 502: The distribution and partial loss.. I think the authors should amplify the concept. To simply define it a "liberation" seems reductive. These are the true signals that consent the dedifferentiation.
- We have edited this sentence
- The remodeling of cell-cell contacts, ECM components and cytoskeleton, allows RPE cells to move out of the layer, then to de-differentiate and change their initial morphology. These changes occur under the influence of “permissive” signals (Fgf, Wnt, IGF1, mTOR, etc.) from the cellular micro-environment and epigenetic signals at the early stages of RPE reprogramming and lead to a change in the RPE cell phenotype into the retinal cell types [65-70].
- line 168: regeneration and show. Remove AND
The sets of DEGs for the early and late stages of regeneration and show the intersection in scores of sets common to both stages.
- We have corrected this misprint
- The sets of DEGs for the early and late stages of regeneration show the intersection in scores of sets common to both stages.
- line 382: AN should be AND
- We have corrected this misprint
- In present work, we have obtained data, indicating that the process of RPE reprogramming in newts P. waltl is accompanied by changes in the transcriptome pattern related to regulation the activity of DNA, RNA, and general protein synthesis, maintenance of genomic integrity, remodeling of specific syntheses, and appearance of specific proteins.

Reviewer 2 Report
Comments and Suggestions for Authors
The manuscript entitled Molecular signatures integral to natural reprogramming in the pigment epithelium cells after retinal detachment in Pleurodeles waltl brings a new perspctive regarding the possibility of neural cells regeneration, having the model of Salamander ability to regenerate their tissues.
Observations
Abstract
Since is an experimental research article, please organize the abstract in Introduction, Material and Methods, Results and Conclusions.
Introduction
- line 58-61 - please reformulate this sentence. What do you mean by "pathology" word? Do you reffer to cellular death due to different diseases you are describing next?
- line 61-63 - please reformulate "scar£ . What is scar tissue? Connective tissue or other kind of tissues (Ex neovascularisation?). Because if the scar tissue contain neovessels as is in hypoxic retina, the regeneration of pigmentar epithelium is not enough to restablish the retina function. Please clarify this aspect.
- line 64-67 - please indicate the references at the end of sentence ; same for the sentence at line 67-72.
In this manuscript there are to long paragraphs with multiple references at the end of the paragraph. I suggest to reconsider this aspect and to put the reference at the end of 1-2 sentences.
Please insert in text the fig 1, before you are presenting it. It is necessary to send the reader to fig 1, and to mention what is fig 1 describing (shortly).
Fig 6 is to small, impossible to be read, even with magnification. Please fix this aspect.ven with references
Conclusions
This chapter is to long and contains again results and discussions, even with references (169).
Your conclusions has to refer strictly at your final results, their importance in this specific field and to the next research perspectives.
Author Response
Dear reviewer. Foremost, we are very grateful for the review, valuable comments made and pointing out the omissions.
Are the conclusions supported by the results? - Must be improved
- We have edited the conclusions to support our results
Comments and Suggestions for Authors
The manuscript entitled Molecular signatures integral to natural reprogramming in the pigment epithelium cells after retinal detachment in Pleurodeles waltl brings a new perspective regarding the possibility of neural cells regeneration, having the model of Salamander ability to regenerate their tissues.
Abstract
Since is an experimental research article, please organize the abstract in Introduction, Material and Methods, Results and Conclusions.
- We has organized the abstract in Introduction, Material and Methods, Results and Conclusions.
Introduction
1. line 58-61 - please reformulate this sentence. What do you mean by "pathology" word? Do you reffer to cellular death due to different diseases you are describing next?
- We have reformulate this sentence
- RPE cell conversion along the mesenchymal pathway (EMT) is considered among the major causes of blindness in humans, including neural retinal detachment from the RPE, proliferative vitreoretinopathy (PVR), age-related macular degeneration [16,18,19].
2. line 61-63 - please reformulate "scar . What is scar tissue? Connective tissue or other kind of tissues (Ex neovascularisation?). Because if the scar tissue contain neovessels as is in hypoxic retina, the regeneration of pigmentar epithelium is not enough to restablish the retina function. Please clarify this aspect.
- We have reformulated the paragraph
- For example, as a complication of rhegmatogenous retinal detachment, RPE cells die and some cells with the acquisition of myofibroblast properties, together with glial cells, form fibrotic epiretinal membrane that is the main cause of PVR and other eye pathologies associated with fibrosis, triggering the neurodegenerative processes [2,20].
3. line 64-67 - please indicate the references at the end of sentence; same for the sentence at line 67-72.
In this manuscript there are to long paragraphs with multiple references at the end of the paragraph. I suggest to reconsider this aspect and to put the reference at the end of 1-2 sentences.
- We have edited the text and placed references in accordance with the cited source:
Studies of RPE cells reprogramming in the newt are stimulated by the assumption that understanding the reprogramming mechanisms will pave the way for the creation of new regenerative technologies for the treatment of RPE-related retinal disorders and for restoring the retinal functions lost as a result of disease or injury in mammals [21, 22]. The ability of RPE cells to change their specialization and become retinal neurons and glial cells in Urodela (thus realizing their retinogenic potential) makes it possible to search for the key molecular factors of initial competence, allowing to overcome EMT and cellular stress, and to achieve a successful implementation of the RPE reprogramming process into neurons in vivo. Studies of newt tissues regeneration are making progress thanks to incorporating highly efficient technologies allow to use micro quantities of tissue for the analysis of transcriptomes, genomes, and proteomes [23,24], as well as for the analysis of genes functions [25,26,27]. Application of these approaches allow to unravel the cellular and molecular mechanisms of ageing, its interplay between regeneration, and contribution of systemic response to injury [28]. The importance of macrophages in facilitating a pro-regenerative environment in the newt eye, helping to resolve fibrosis, and modulating the overall inflammatory landscape have been proved [29].
4. Please insert in text the fig 1, before you are presenting it. It is necessary to send the reader to fig 1, and to mention what is fig 1 describing (shortly).
- Figure 1 is inserted in the text after it has been presented:
In the present study, we have focused on transcriptome analysis of the initial stages of RPE cells reprogramming, using a model of experimental mechanical retinal detachment in Pleurodeles waltl (Figure 1).
5. fig 6 is to small, impossible to be read, even with magnification. Please fix this aspect.ven with references
- Figure 6 has been replaced with a clear one
Conclusions
This chapter is to long and contains again results and discussions, even with references (169).
Your conclusions have to refer strictly at your final results, their importance in this specific field and to the next research perspectives.
- We have edited the conclusions
